## [Editor Report · Decision Letter 0]

4 Feb 2025

Dear Arash and David,

Thank you for submitting your manuscript entitled "Translocation of gut commensal bacteria to the brain" for consideration as a Discovery Report by PLOS Biology.

Your manuscript has now been evaluated by the PLOS Biology editorial staff, as well as by an academic editor with relevant expertise, and I am writing to let you know that we would like to send your submission out for external peer review. I have discussed the manuscript with the Academic Editor, and we would send it out to review as it is as Discovery Report.

Once your full submission is complete, your paper will undergo a series of checks in preparation for peer review. After your manuscript has passed the checks it will be sent out for review. To provide the metadata for your submission, please Login to Editorial Manager (https://www.editorialmanager.com/pbiology) within two working days, i.e. by Feb 06 2025 11:59PM.

Best wishes,

Melissa

Melissa Vazquez Hernandez, Ph.D.

Associate Editor

PLOS Biology

---

## [Decision Letter · Decision Letter 1]

22 Apr 2025

Dear Arash,

Thank you for your patience while your manuscript "Translocation of gut commensal bacteria to the brain" was peer-reviewed at PLOS Biology. Your manuscript has been evaluated by the PLOS Biology editors, an Academic Editor with relevant expertise, and by three independent reviewers. I would like to sincerely apologize for the extremely long time to get back to you with a decision.

As you will see in the reviewer reports (which are really thorough), although the reviewers acknowledge the potential interest in your findings, they have also raised a substantial number of crucial concerns, and we think they have offered really important advice that should be followed. Based on their specific comments and following discussion with the Academic Editor, it is clear that a substantial amount of work would be required to meet the criteria for publication in PLOS Biology. Given our and the reviewer interest in your study, we would be open to inviting a comprehensive revision of the work; however, this would need to thoroughly address all the reviewers' comments in full. We think that the study can be ground breaking but extraordinary findings require extraordinary evidence.

In the reports, you'll see that Reviewer 1 mentions multiple concerns such as a lack of details on the methodology that makes it hard to evaluate the results, and that there is also some overinterpretation of results in the text not matching the significance shown in the figures. This reviewer also questions that the reason of seeing bacteria in the brain is solely due to intestinal permeability, that the bacteria in the brain are commensals, and the reason of leaving out some experiments that were in your preprint and that would have supported some of their claims. Reviewer 2 mentions several methodological and interpretation issues. Some concerns mentioned are the generalizability of the findings, a mechanistic explanation of how the bacteria is getting to the brain (if it is through blood, how is it being cleared), the quantification methods. This reviewer recommends to use qPCR to assess microbial translocation to the blood, and requests that the you explore bacterial clearance in the brain and to use imaging techniques to evaluate the route of translocation. The reviewer also recommends you to do vagotomy and evaluate barrier breaching. Reviewer 3 has several concerns that would require clarification and additional experiments. The reviewer wonders if there might be transient bacteremia that you may not be seeing, and recommends qPCR, FISH and measuring antimicrobial peptides to evaluate this. The reviewer also asks if the bacteria in the brain is intracellular and requests that you analyse the intestinal barrier.

Although Reviewer 2 and Reviewer 3 suggest using DNA based approaches such as 16S qPCR or metagenomics to verify their results, which is in principle fine, the reported bacterial biomass is so low in many of these samples that it may well be below the reliable detectable limit of these methods. Reviewer 1, an expert in low biomass studies, considers it likely that you find that contamination will swamp out any true signal, which might actually make the situation more muddled, not less! If such DNA-based approaches are going to be done, then they have to be done extremely stringently. Please see Box 1 in this paper for more thoughts on "low/no biomass" samples: https://www.nature.com/articles/s41586-022-05546-8. If you do 16S-based qPCR, you should do the spiking experiments mentioned in Box 1 to accurately determine their limit of detection with this approach. As suggested by R3, FISH would probably be worth trying, although we acknowledge it may prove difficult to detect bacteria in the samples if they are only present as a few cells in the entire brain/vagus nerve.

The Academic Editor has also said "they can sequence the PCR product to see if it matches the bacteria in the brain. That would be another way around the contamination issue. A contaminant from the reagents wouldn't match." While this is not a requirement, we hope it may be useful as a trouble-shooting tip.

We appreciate that these requests represent a great deal of extra work, and we are willing to relax our standard revision time to allow you 6 months to revise your study. Please email us (plosbiology@plos.org) if you have any questions or concerns, or envision needing a (short) extension.

**IMPORTANT - SUBMITTING YOUR REVISION**

*Resubmission Checklist*

*Published Peer Review*

*PLOS Data Policy*

*Blot and Gel Data Policy*

Sincerely,

Melissa

Melissa Vazquez Hernandez, Ph.D.

Associate Editor

PLOS Biology

REVIEWERS' COMMENTS:

Reviewer #1:

This is quite an intriguing paper. The finding that, given the required genetic background and/or host diet, bacteria can translocate from the gut of mice to the brain via the vagus nerve is potentially of widespread interest. It is clear that the authors have carried out a large amount of work, and provide some quite compelling evidence.

That being said, I did find some significant issues with the paper. I felt that interpretations were sometimes prone to over-extrapolation, some of the results were a little puzzling, and the manuscript lacked sufficient detail in places to allow thorough assessment of the robustness of certain conclusions. Given that the authors are making some big (and potentially dogma changing) claims here, I therefore think that a significant amount of revision is required to ensure accuracy and robustness. I apologize for the length of comments to follow, but hope that these will be helpful for the authors when revising.

Major comments:

1. There is a lack of detail on how some of the results were generated, which unfortunately makes it difficult to assess the robustness of conclusions. As an example, it was not explained how "CFU/g" was calculated for the brain samples. Looking at the relevant methods section it states on line 393 that "150-500 uL" of homogenate was plated. However, to work out accurate CFU/g the authors would need to know exactly how much homogenate was added to each plate? Was this factored into the CFU/g calculations? The other issue with presenting results as CFU/g is that the mouse brain typically weighs significantly less than 1 gram (average of around 0.4 to 0.5 g?). This means that, when extrapolating some of the CFU/g results to the whole brain of mice, then the actual numbers of CFUs must be very low. For example, in cases where less than 10 CFUs/g were reported (e.g. see Fig 2F), when you divide that figure by the total mouse brain weight of around 0.5 g then you are effectively talking less than 4 or 5 CFUs in the entire mouse brain, sometimes less than 1 CFU? Can CFU per gram be reliably calculated in cases when you only get an occasional colony or two on an agar plate? More detail in the Methods section would help to clarify. Regardless, I think it might also be helpful to add the actual number of colonies counted from brain homogenates as a supplemental info table, rather than just convert to CFU/g?

Further examples where insufficient detail was provided are given in the "Specific comments" below.

2. Some of the results are presented in quite an anecdotal way. There appear to be numerous results discussed as being noteworthy/significant in the main text where there was no significant difference according to the figures (out of the "brain bacteria" species mentioned, only S. xylosus and E. cloacae were found to be significantly different, the results for S. sciuri, E. faecalis and P. cineris appear to be non-significant - see Figs S3D, S3H and S4D). Please see specific comments below for more details.

3. A central premise is that increased intestinal permeability allows bacterial translocation to the vagus nerve. However, this premise is not necessarily entirely supported by the results presented. If you look at the data in Figures 1, 2 and 3, average intestinal permeability seems to be somewhat higher in the B6 mice (see Figs 2D and 3D - average of 4 or 5, with outliers up to around 14 ug/ml) vs the Mdr2-/- mice (see Figs 1B and 3A - average of 1 or 2, with outliers up to around 4 ug/ml). However, the CFU/g numbers are actually higher in the brains of Mdr2-/- mice (average of around 10^2 in Figs 1E and 3C) than in the B6 (average of around 10 or less in Figs 2F and 3E), despite the intestinal permeability apparently being lower in the Mdr2-/- mice. If increased intestinal permeability allowed increased translocation, then the opposite pattern should be observed?

4. The claim that the microbes travel to the brain via the vagus nerve is an intriguing hypothesis. When reading the article, it occurred to me that one potential way to test this hypothesis would be to perform vagotomy on the animals. On researching this further, I came across the bioRxiv preprint of this article (https://www.biorxiv.org/content/10.1101/2023.08.30.555630v1.full), which showed that the authors had indeed carried out partial vagotomy, and the results seemingly supported their hypothesis. I therefore wondered why those results had been removed from the current version of the manuscript? Were they not robust?

5. Similarly, another question I had upon reading the article was whether or not presence of these microbes in the brain had any impact on host animal physiology/health. On reading the bioRxiv preprint, it seems that the authors had previously demonstrated that it was linked with neuroinflammation and neural protein aggregation. I wondered why these data were also removed from the current version of the manuscript? Were these data also not robust?

6. I am not convinced that the bacteria that they detect in the brain are truly "gut commensals". The gut is not the natural habitat of some of the species they detected in the brain (more on this below). Furthermore, "commensal", by definition, means that they do the host no harm, but in most cases they have identified organisms that have been shown previously to be capable of pathogenesis, including infections of the brain in KO mouse models. Looking first at the example of S. xylosus, this is most commonly a skin microbiota organisms, not normally dominant in the gut. There is also prior evidence that this species can cause deep tissue infections (including of the brain) in KO mouse models (see https://pmc.ncbi.nlm.nih.gov/articles/PMC2919191/, and https://pmc.ncbi.nlm.nih.gov/articles/PMC5557206/). Calling this a "gut commensal" therefore seems like a stretch. The same applies to S. sciuri, which is also predominantly a skin microbiota species with previous evidence for deeper infections in mice, humans and other animals (https://pubmed.ncbi.nlm.nih.gov/36044992/ ; https://pmc.ncbi.nlm.nih.gov/articles/PMC1151920/ ; https://pmc.ncbi.nlm.nih.gov/articles/PMC1764720/). Enterobacter cloacae and Enterococcus faecalis, although gut-dwelling microbes, are similarly relatively common causes of clinical infections. Paenibacillus cineris is an environmental microbe, not a gut-dwelling one, and its presence in the mouse model here seems largely to be a result of broad spectrum antibiotic treatment. Taken together, I am therefore not convinced that any of the named species meet both the "gut" and "commensal" status that is repeatedly claimed throughout the manuscript. It could therefore be claimed that what is being observed are infections rather than incidental translocation by harmless gut microbes? Infection of pathogens via the vague nerve would in itself be interesting, of course, but that is a rather different interpretation than what the authors currently present.

7. I felt that the Discussion was lacking in depth analysis of many key features of the text. There is also not a single reference to any other piece of published research in this whole section. I would have expected to see some coverage of previous evidence for deep tissue infections by the named "brain bacteria" (see examples in the comment above), details of what mechanisms other microbes (e.g. numerous viruses, and potentially bacteria like B. burgdorferi) have previously been shown to use in order to infect and travel along the vagus nerve, and how high fat diets can increase intestinal permeability. Some discussion of the potential relevance of the mouse models for humans would also be useful. For example, how does the composition of the Paigen diet compare to typical "Western" diets? What is the evidence for similar intestinal permeability in patients with neurodegenerative disorders? Have bacteria reproducibly been detected in brain samples from autopsies of patients with these conditions? If so, are they the same sort of species as those that were detected in this study, and is there any evidence that this is linked with diet? This sort of additional context seems important if the authors wish to make the case that their findings in these relatively contrived mouse models have relevance for humans?

Specific comments (in order they appeared in the text):

Line 1 - The title is currently too non-specific, and therefore open to misinterpretation and over-extrapolation. Aside from my previous concern (see comment 6 above) that the authors are mostly not seeing translocation of "gut commensals", they also need to make clear that the work was done (mostly) in KO mouse models. I suggest that something more like "Translocation of bacteria from the gut to the brain of mice fed a high fat diet, and in murine models of neurodegenerative diseases" would more accurately reflect the content of the article.

Line 62 (and all other similar subsequent uses throughout the rest of the article) - as outlined in comment 6 above, I am not sure it can be claimed that the species they detect in the brain are truly gut commensals, so they probably should not be referred to as such throughout.

Lines 66 to 74 - the Introduction is rather short, and lacking in key background content information on topics like links between high fat diets and intestinal permeability, and transneuronal infections, but I guess this is not so important if these are dealt with in a revised Discussion instead (see comment 7 above)

Line 85 (and all other subsequent uses in the text) - a minor point, but "Lactobacilli" is not the formal taxonomic name ("Lactobacillus" is), so it should not be in italics, or have a capitalized first letter. This also applies to similar later uses of the words "Staphylococci" and "Enterococci" (see lines 100 and 131), which should also not be in italics or have a capital first letter.

Line 87 - were the authors able to tell from the 16S rRNA gene sequencing if these Staphylococcaceae were S. xylosus or S. sciuri?

Lines 100 to 101, 114 to 115, Figs 1G to 1I, Fig S3 and Fig S4 - How were CFUs of S. xylosus, S. sciuri, E. faecalis and P. cineris per gram of feces and ileum contents calculated? I couldn't see this information in the Methods section. For the brain, it seemed that the relatively small number of colonies from brain homogenates were picked and then identified via MALDI-ToF, and then this could potentially be extrapolated to CFU/g, but how was this done with fecal and ileum samples? Were colonies that grew from feces and ileum samples also routinely identified via MALDI-ToF? These sample types would of course include many different types of different gut bacteria. To get accurate final figures of between 10^6 and 10^9 (e.g. see Fig 1G) for specific species like S. xylosus then a very large number of colonies would need to have been picked and identified in this way, which seems expensive/infeasible at scale? Please add full details to the relevant Methods section explaining how S. xylosus, S. sciuri, E. faecalis and P. cineris were accurately enumerated in sample types other than brain tissue.

Line 105 - "S2 Fig" should be "S2 Fig G-I"?

Line 110 (and all other subsequent similar uses throughout the manuscript, including figure legends in supporting material) - another minor, pedantic, point (sorry) but 16S rRNA gene sequencing is neither genomic or metagenomic as it focusses solely on one gene rather than whole genomes. Please therefore amend all uses of "16S genomics" and "16S metagenomics" to something like "16S rRNA gene profiling".

Line 119 - I was confused by the "Data presented from minimum 3 independent experiments (n=3-5/group)" text here. There are only 6 data points for control in Fig 1J, and 8 for Paigen in both Figs 1K and 1J, so I wasn't sure how this agreed with 3 experiments with at least 3 per group (minimum number of data points should be at least 9)?

Lines 129 to 131 - in the case of both E. faecalis and S. sciuri this association was not significant (see Fig S3), so the text could probably be edited here to improve accuracy/robustness?

Line 135 - could clarify the mouse background here. Were these also Mdr2-/- mice?

Line 139 - I wondered where P. cineris came from? This seems to be an environmental species, not a gut dweller? It is also potentially surprising to see it invading the vagus nerve (albeit, the results shown in Fig S4D seem to be non-significant?) since I could only find one case report of P. cineris acting as a pathogen (in a CF patient with pneumonia - https://www.medrxiv.org/content/10.1101/2023.09.19.23295794v1.full). It has obviously benefitted from the antibiotic treatment killing off competing species within the gut, but does this species have the necessary genomic repertoire that allows it to effectively colonise mammalian hosts and invade deeper tissues? At the very least it might be nice to have some text on this added to the Discussion section?

Lines 139 to 142 - this correlation is not significant according to Fig S4D?

Line 141 (and all other subsequent similar uses throughout the manuscript, including figure legends and table column headers in supporting material) - another minor pedantic point (sorry) but "sp." (or "spp.") should not be in italics.

Line 148 - add "in Mdr2-/- mice" to the end of the sentence here to improve clarity?

Lines 152 to 154 - According to the figure legend, Figs 2A, 2B and 2C appear to show "CFU of bacteria" rather than CFUs of S. xylosus and E. faecalis? It was therefore not clear how the data in those figures allowed the authors to make the claim that they detected these specific species.

Lines 159 to 160 - "some mice" is a little non-specific and anecdotal. Looking at the data in Fig2E, for example, it looks like only one CFU was detected in only one of the mice tested at day 2, all the others were negative. Results were also non-significant according to Fig 2E, at all time points, including days 2 and 4.

Lines 160 to 161 - As above, this was anecdotal. Bacteria were not detected in the vagus nerve of most mice, and the results were non-significant according to Fig 2E. Furthermore, Fig 2F shows an average of perhaps 3 or 4 CFU/g of brain. Given the average mouse brain weight of 0.4 to 0.5g, this equates to around 1 or 2 CFUs per entire brain?

Lines 163 to 165 - This claim is also anecdotal. According to both Fig 2E and Fig S5 none of the vagus nerve-related findings were significant, therefore the claim that "bacteria were detected in the vagus nerve prior to detection in the brain" is not supported by the available data?

Lines 172 to 173 - I wondered if the variation observed here might be informative? Was there any difference between the 40% of mice where E. cloacae was cultured from brains to the 60% that weren't? Were they housed in different cages, sampled at different times indicating batch effects, or did the 40% harbor greater numbers of E. cloacae in their guts, or have greater intestinal permeability, than the other 60%, therefore increasing likelihood of translocation?

Lines 173 to 175 - Did the authors also monitor E. cloacae levels in blood and vagus nerve samples (only fecal pellet, ileum and brain results are presented)? Given that E. cloacae can cause bacteremia/sepsis (e.g. https://www.sciencedirect.com/science/article/pii/S2666524721000987 ; https://pmc.ncbi.nlm.nih.gov/articles/PMC10464967/ ; https://www.sciencedirect.com/science/article/pii/S2214250924000945), it is plausible that this particular species might have entered the brain via the bloodstream rather than the vagus nerve?

Line 181 - "Fig J, K" should be "Fig2 J, K"?

Lines 181 to 183 - According to Fig 2L, this result was non-significant, so this section is anecdotal.

Lines 222 to 238 - Looking at the supplemental metadata xls file, it looks like only 5 mice per group were studied? This seems underpowered? Also, how were cage effects controlled for? I could not see any information about this in the Methods section. Can you be sure that the observed differences were due to the genetic background rather than to housing conditions? It is well established that co-housing/cage effects and coprophagy can be a far bigger driver of overall microbiota composition than genetics, which can severely confound comparative rodent microbiota studies (see, for example, the "Cage effects in animal experiments" section in the following article: https://microbiomejournal.biomedcentral.com/articles/10.1186/s40168-017-0267-5).

Line 231 - perhaps clarify here what mouse background the WT group was (B6?).

Line 231 - add "(Fig 4A and S9 Fig)" after the word "mice"?

Line 233 (and Fig 4A) - "Coriobacteria" should be "Coriobacteriia"?

Line 233 - "S9 Fig" should be "S10 Fig"? S9 seems to be showing the APP results, not LRRK2?

Lines 241 to 243 - as with previous comment above (on Lines 100 to 101, 114 to 115, Figs 1G to 1I, Fig S3 and Fig S4), it was not clear to me how specific species of bacteria in fecal samples were detected and described. Please can this be clarified, and additional information added to the Methods section?

Line 258 - Looking at the supplemental metadata xls file, it looks like only 5 mice per group were included for microbiota composition analysis? I was therefore a bit confused by the "Data presented from minimum 2 independent experiments (n=3-5/group)" text here. How can the data presented in Fig 4A, which seems to be based on 5 mice per group, be based on 2 experiments with at least 3 per group (minimum number of samples should be at least 6)?

Lines 272 to 274 - I'm not sure this statement is true? Please see my earlier major comment number 3.

Lines 295 to 300 - Swabs were tested for contamination, but the amount of material on those swabs is likely to be less than that in some whole organ homogenates? Low level contamination might therefore have been missed on swabs, but picked up from larger samples?

Lines 327 to 328 - "In several models" should be "In two models" (Mdr2-/- and B6)?

Lines 334 to 349 - as discussed above, this section should include some text on animal housing arrangements (how many animals per cage etc), and what measures were taken to control for cage effects.

Lines 374 to 382 - This is obviously a different method to that used to monitor CFUs in organs. Only 100 ul of blood was sampled, vs entire organs, which in the case of the brain would likely be more than the equivalent input of 100 ul?. Given the very low colonisation numbers in many samples I therefore wondered if there was a risk of "missing" some bacteria in blood vs organs due to the lower amount of initial collected material?

Lines 391 to 393 - This is not necessarily the most stringent method of sterilization (heat killing would be more effective)? Were samples processed together in batches? I note that it says brains were dissected first, but was there any indication of batch effects that might result from low level transfer of live bacteria after dissection of gut organs from one animal then moving on to the brain of another?

Line 393 - as discussed previously in major comment number 1, was the actual amount of homogenate that was added to agar plates recorded for every sample? This would seem to be essential information for calculating CFU/g? As mentioned previously, adding information on how CFU/g was calculated to this section is important. I also had concerns that the accuracy of these CFU/g figures might be impacted by extrapolating from a very small number of CFUs on a plate.

Line 401 - was the same batch of PBS used for both control and Paigen fed mice? Was PBS plated out to ensure there no CFU growth?

Lines 412 to 415 - what method was used to detect E. cloacae on day 5 after gavage? Similar to the earlier comment (on lines 100 to 101, 114 to 115, Figs 1G to 1I, Fig S3 and Fig S4) it would be good to know how the abundance of this specific species in samples was accurately monitored? The text on 413 to 415 indicated MALDI-ToF was used, but how many colonies were picked, and how was CFU/g of feces/ileum accurately calculated?

Lines 424 to 426, and line 429 - what culturing method was used, and what PCR primers etc? There is not enough detail here to allow independent verification?

Line 432 - should be "16S rRNA gene sequencing" since the gene is sequenced, not the actual rRNA?

Lines 451 to 454 - some of this is not relevant, since these alpha diversity measures are not presented in the paper?

Line 455 - "based relative" should be "based on relative"?

Lines 456 to 458 - Were p values corrected to allow for the false discovery rate following multiple comparisons? If so, what method was used for correction? If not, why not?

Line 460 - LEfSe seems to have been used to generate the results and figures shown Fig 1A, Fig 4A, and Supp Figs 1A, 9, 10 and 11? If so, details on this should be added here, and a reference for the software added?

Line 467 - "illumine" should be "Illumina"?

Lines 481 to 483 - this was potentially a little vague, and I'm not sure I fully understood what data and materials could be obtained and how. Perhaps this could be expanded slightly to clarify?

S2 Fig - Were these tested for significance (this is not indicated in the figure legend). If so, presumably none of the results were significantly different? This could be added to the figure legend for clarity?

The legend also states that "Data are representative of 3 independent experiments". How many samples were included per experiment? Was there any batch variation?

Fig S3H - there were only detectable CFU/g in two out of all mouse brains tested (the rest were negative), and the number of CFU/g looks like it is only around 10? As discussed in comment number 1 above, when then converting back into number of CFUs per whole mouse brain, this would be less than 5 per brain? How many actual CFUs were observed on agar plates?

S5 Fig - How were CFU/g calculated for vagus nerve samples? The results here indicate that CFU/g were pretty much always less than 10 CFU/g. How much does the vagus nerve weigh? Presumably not very much, and far less than 1 gram? So I was not sure how you end up with an accurate enumeration of 1 CFU per gram (as seems to be the case for three of the vagus nerve samples). That must equate to much less than 1 CFU per entire vagus nerve? How many CFUs were observed on the actual plates, and how was CFU/g calculated from this?

Table S1, legend - Were the streaked colonies confirmed to be S. xylosus or Enterobacter sp.? If so, how so?

Table S2 - I found this quite confusing, sorry. What was meant by "Data are representative of two biological representatives each"? How does this relate to the numbers in the "Isolates" and "Number of isolates" columns? How many gut-derived isolates of each species were there, and how many brain-derived. And how many of each were sequenced? None of this is immediately intuitive (at least to me) from the table.

Table S3 - are the figures here accurate? For WSB.APP brain the table shows 14/15, but Fig 4C indicates 15/15? For LRRK2 brain it shows 12/19, but Fig 4C indicates 10/15? For BTBR vagus nerve it shows 4/8, but Fig 4D indicates 8/13?

Furthermore, it is also not clear what was meant by "Data are representative of two biological representatives" in the legend at the bottom of the table. Please could this be clarified?

Table S4 - are the figures here accurate? For WSB.APP brain the table shows 14/15, but Fig 4C indicates 15/15? For LRRK2 brain it shows 12/19, but Fig 4C indicates 10/15?

"parkinson's" in the table should also be "Parkinson's"?

Finally, could the authors please clarify what was meant by "the bacterial colonies found in the brain were similar to those found in the same mice's gut"?

Reviewer #2:

The study presents an innovative concept by demonstrating that dietary-induced gut dysbiosis (via a Paigen diet) can lead to the translocation of commensal gut microbes to the brain via the vagus nerve. The work has potential to bridge environmental (dietary) and genetic factors in the etiology of neurological disorders. However, several methodological and interpretative issues need to be addressed here, before the study can be seriously considered for publication.

Major comments:

Introduction and insufficient background: The introduction is unusually concise and does not adequately discuss the significant literature on gut dysbiosis and its association with neurodegeneration. A more thorough review of prior findings on indirect mechanisms linking the gut and brain is warranted.

Contextual gap: Additional discussion regarding how current findings build on or diverge from existing knowledge would greatly enhance the reader's understanding of the study's novelty.

Choice of animal model: The use of a cholestatic liver disease mouse model (MDR2 knockout) is not sufficiently justified. The authors should explain why this model was chosen over standard mouse models and whether the observed phenomenon is limited to this specific genotype. This loops back to the poor-introduction and background comment as well.

Generalizability: Clarification is needed on whether the translocation process observed is unique to the MDR2 knockout model or can be generalized to other models, including wild-type mice.

Intestinal permeability and bacterial translocation: Despite a significant increase in intestinal permeability under the Paigen diet, there is no evidence of bacterial translocation into the blood. The authors should provide a mechanistic explanation or alternative hypothesis (e.g., rapid immune clearance or sequestration of bacteria) to reconcile these findings.

Immune clearance: It is essential to determine if enhanced immune activity (such as elevated antimicrobial peptides) is responsible for clearing bacteria from the blood. Including data on systemic antimicrobial responses would strengthen the conclusions.

Perhaps a more basic approach of QPCR based quantification of 16S rRNA would be highly recommended to assess the microbial translocation to the blood (which could be intracellular in nature).

Methodology for bacterial quantification: The CFU/g values presented in fig. 1 are obtained via MALDI-TOF, a method primarily suited for qualitative identification. A qPCR-based approach or other quantitative methodologies would be ideal to validate the CFU based quantitative findings.

The similarity in CFU values obtained in per gram of stool and per gram of ileum is intriguing (fig 1 C/D/E). The authors should discuss potential reasons for this observation and ensure that sampling and quantification techniques are appropriately validated across different tissue/sample types.

Dynamics of bacterial presence in the brain: In fig. 3 (panels D & E), intestinal permeability decreases sharply at 7-14 days while CFU/g in the brain remains elevated until 14-21 days before declining at day 28. This transient nature of microbial presence in the brain is intriguing The authors need to explore whether this pattern reflects delayed bacterial clearance, limited colonization capacity in the brain, or other host-mediated effects.

A discussion regarding whether the brain environment is inherently non-conducive to sustained microbial survival or if active clearance mechanisms (such as microglial responses) are at play would provide valuable mechanistic insights.

Route of translocation: While CFU analyses suggest the vagus nerve as a potential conduit for bacterial translocation, direct imaging evidence would significantly strengthen this claim. The authors should consider incorporating microbial imaging techniques to visualize the entry, transit, and exit points of the microbes along the vagus nerve.

Intracellular vs. extracellular localization: Determining whether the bacteria is translocated and resides intracellularly or extracellularly in the brain could provide clues about the mechanism of translocation and potential immune interactions. Studies have shown bacterial translocation across tissues in monocytes or macrophages, that possibility should be explored though simple imaging.

Impact of diet on fecal parameters: The manuscript does not address differences in stool frequency, weight, or moisture content between Paigen diet-fed and control mice. Given that high-fat diets can influence stool parameters and intestinal transit time, these factors should be measured and discussed as potential confounders.

Neurological disease models: The study extends its findings to neurological disorder models, yet it remains unclear whether whole microbiota translocation is observed across these models. Detailed quantification of total CFU in brain tissue, beyond specific organisms like S. xylosus, is necessary to generalize the findings.

Given that the data suggest barrier breach is pivotal for microbial translocation to the brain (Fig. 4D/E), it is essential that the authors provide a detailed characterization of the barrier disruption induced by the Paigen diet. Moreover since this breach in barrier is not affecting translocation to blood, its spatial and cellular characteristics must be determined in Paigen fed mice. It would be advisable to incorporate imaging analyses—such as mucin staining in the ileum and immunolocalization of tight junction proteins—to map and quantify the extent of the barrier compromise. This detailed evaluation would not only validate that the Paigen diet primarily mediates microbial translocation through barrier breach but also strengthen the overall mechanistic link between dietary-induced gut dysbiosis and neuroinflammation.

Broad comments:

Mechanistic studies on vagal involvement: Experiments involving pharmacological or surgical interventions (e.g., vagotomy) in mice to more definitively establish the role of the vagus nerve in microbial translocation would be ideal. I strongly recommend this if the claim is to be made for 'vagus nerve mediated translocation to brain'.

Longitudinal studies: Considering a longer observation period could help determine whether the transient appearance of bacteria in the brain is a common outcome or if there are potential long-term effects on neurological function if microbes remain for long period.

Reviewer #3 (Archita Mishra):

This manuscript examines whether a high-fat (Paigen) diet can induce gut dysbiosis that facilitates the direct translocation of commensal bacteria to the brain, potentially via the vagus nerve. The study aims to bridge environmental factors with neuroinflammatory processes implicated in neurological disorders. It introduces an interesting link between gut barrier integrity under a dietary stressor and neuroinflammatory processes, suggesting that the vagus nerve may act as a conduit—this is an exciting perspective that can shift current paradigms.

While the work introduces an interesting concept, there are several areas that require additional clarification and experimental support.

The introduction currently lacks a comprehensive review of the literature on the gut-brain axis. A more detailed discussion of background, showing and fully acknowledging the previous works on similar lines is recommended. Comparing indirect translocations (e.g., bacterial metabolites) with the direct translocation hypothesis would provide useful context.

Using microbial barcoding to trace specific gut-derived bacteria within the brain is innovative—a concept that holds significant potential in elucidating the mechanisms of gut-brain communication. However, this promising experiment is inadequately presented, as the key data reside mostly in supplementary tables rather than being integrated into the main figures and discussion. This omission weakens what could be a critical piece of evidence demonstrating the direct translocation pathway.

Mechanistic insights into translocation and clearance: The absence of bacteria in the bloodstream, despite increased gut permeability, prompts questions about transient bacteremia and systemic immune responses. Employing highly sensitive techniques (e.g., 16S rRNA qPCR) for detecting low-level bacterial DNA in blood or circulating phagocytes would help resolve this ambiguity. A characterization of host immune responses, perhaps through cytokine profiling or analyses of antimicrobial peptide levels, is needed to determine whether enhanced immune clearance selectively confines bacterial migration to a neural route.

The rationale for using the Mdr2-/- mouse model should be more clearly justified. How does this model, which is primarily used to study cholestatic liver disease, relate to the mechanisms of bacterial migration?

Quantitative and imaging analyses: The use of CFU counts and MALDI-TOF for bacterial identification provides a basic level of evidence; however, incorporation of molecular methods (such as 16S rRNA gene qPCR, FISH, or metagenomic approaches) would strengthen the quantitative assessment of bacterial loads in different tissues.

Direct imaging evidence to support the role of the vagus nerve as a conduit for bacterial transit is lacking. Advanced imaging methods, for example confocal microscopy of the nerve and brain with specific bacterial markers, would significantly enhance the mechanistic claims.

Temporal dynamics and localization in the brain: The manuscript notes that bacterial load in the brain remains elevated even as gut permeability decreases. A systematic time-course study that correlates bacterial presence with markers of neuroinflammation (e.g., microglial activation) would help clarify whether the brain environment is actively clearing bacteria or if there is sustained colonization.

It is also important to establish whether bacteria are localized intracellularly within resident phagocytic cells or remain extracellular, as this distinction could have implications for the ensuing neuroinflammatory response.

Intestinal barrier function and confounding variables: Analyses of the intestinal barrier, including assessment of tight junction proteins and mucins, are necessary to support the claim that the Paigen diet compromises gut integrity in a way that facilitates bacterial translocation (reduction of barrier proteins).

Other gastrointestinal parameters that are influenced by high-fat diets, such as stool consistency, transit time, and local immune responses, should be measured and discussed as potential confounding factors.

Extension to neurological disease models: The application of these findings to models of Alzheimer's, Parkinson's, and autism spectrum disorder is of interest. However, a more thorough quantification of overall bacterial presence in different regions of the brain is desired. What cell types in the brain largely come in contact with bacteria is a question worth asking here.

Contamination controls and tissue processing: Given the unsuspected finding of bacteria in the brain, the authors should provide a more thorough description of tissue collection and processing procedures. Detailed protocols that minimize the possibility of exogenous contamination during dissection and sample processing should be clearly outlined.

In summary, this manuscript addresses an important aspect of the gut-brain relationship by suggesting a direct microbial translocation pathway under conditions of dietary stress. To improve the study's rigor and impact, it is recommended that the authors expand their literature review, provide additional mechanistic data (especially regarding bacterial imaging, direct evidence of translocation and barrier integrity), and enhance their quantitative and imaging approaches. Addressing these issues will help ensure that the conclusions drawn are well-supported and that the potential implications for neurodegenerative disease research are clearly established.

---

## [Decision Letter · Decision Letter 2]

29 Oct 2025

Dear Arash,

Thank you for your patience while we considered your revised manuscript "Translocation of gut bacteria to the brain" for consideration as a Discovery Report at PLOS Biology. Your revised study has now been evaluated by the PLOS Biology editors, the Academic Editor and the original reviewers.

As you will see in the reports, the reviewers are generally positive about the revision, with Reviewers 2 and 3 satisfied. Reviewer 1 is convinced about your findings. However, R1 raises important concerns about the framing and interpretation of the data. In particular, they note that the quantitative analyses based on CFU counts may not be statistically robust, and therefore the data should be treated as qualitative. They also highlight that some conclusions remain overstated, the terminology could mislead readers, and that several previous issues (such as clarity on non-significant results, cage/batch effects, and methodological details) were not fully addressed in the manuscript.

After discussion among the editorial team and the Academic Editor, we agree that these points are valid and important to resolve. As the findings could be easily misinterpreted, we believe that following Reviewer 1’s recommendations will strengthen the paper and ensure balanced interpretation without diminishing its significance. We therefore ask that you revise the manuscript accordingly before we can proceed further.

**IMPORTANT - SUBMITTING YOUR REVISION**

*Resubmission Checklist*

*Published Peer Review*

*PLOS Data Policy*

*Blot and Gel Data Policy*

Sincerely,

Melissa

Melissa Vazquez Hernandez, Ph.D.

Associate Editor

PLOS Biology

REVIEWERS' COMMENTS

Reviewer #1:

I thank the authors for the significant and impressive amount of work they have done since the last submission. Overall, I am now convinced that the central finding, i.e. that certain bacteria can translocate in small numbers from the gut to the brain via the vagus nerve in these mouse models, is probably correct. The authors have been admirably rigorous in their lab work. However, in my opinion they have unfortunately not always applied the same scientific rigour to their interpretation and presentation of the results, which contain over-extrapolations and selective reporting. They have also paid lip service to many of the concerns raised previously by peer reviewers in their rebuttal letter, but not actually dealt with all of them appropriately in the main text.

Therefore, I'm afraid I do not recommend publication in its current form, and request significant further edits to the text. I have included detailed comments below, and hope that these will be useful for the authors, and help to ensure the robustness of the published work.

Major comments:

1. I thank the authors for providing the previously requested colony count data as supplemental info (although note that this is not complete, as they do not provide the counts that underpin the data shown in Figs 1G, 1H, 1I, 1K, 1L, 2E, 3D, S2H, S2I, S3A-H, S5A, S5E, S5F, S9, S10B, S17B, S17C, S17D and S17E. Furthermore, the fraction figures are an odd way of presenting the data, they should instead produce the direct counts from the actual dilutions they were counted from).

However, this data raises serious concerns about the robustness of the calculated CFU/g figures. It has long been established in microbiology that around 30 to 300 colonies per plate are required for accurate CFU/g or CFU/ml figures. Lower than that and the results can be heavily impacted by slight differences in colony count, higher than that and the numbers are too high for accurate counting (for a short historical perspective on this, please see https://pubmed.ncbi.nlm.nih.gov/27815539/. Of particular note, please see the quote "it is at once clear that plates having less than 20 and more than 400 colonies are so apt to be widely discrepant that counts from plates of this sort should be disregarded.") When looking at the actual count data it becomes clear that a large proportion of the CFU/g results (particularly those from the vagus nerve and brain samples) fall far outside of the statistically robust range. For Figure 5E, as an illustrative example, many of the results are based on a single colony being detected. In contrast, it seems some data for Fig S6 C Ileum and S6 D brain included CFU/g counts of over 1000 colonies per plate?

The unfortunate consequence of this is that much of the subsequent statistical comparisons of CFU/g results, for example with the Mann Whitney testing, is not based on robust count data. Therefore, most of these statistical comparisons are invalid. To be clear, this does not mean translocation is not happening, but it does mean that their data are in general qualitative, not quantitative, and the results should therefore be presented throughout as such. This will require significant revision of figures and associated text, but will not need any additional experimental work in the lab.

2. The Discussion is still too one-sided, mostly repeating results and attempting to convince the reader rather than critically assessing the observations. In particular, the description throughout the whole main text of these translocated bacteria as "gut bacteria" has the potential to be significantly misinterpreted and/or over-extrapolated by new readers. Out of the thousands of species that can colonise the gut, only a very small number of specific species are being detected in the brain here. Importantly, and as discussed in my earlier peer review comments, all of the species they repeatedly find in the brain are opportunistic pathogens, some of which have prior evidence for causing deep tissue infection, including of the brain, in KO mouse models (please see original peer review comments for examples from the literature that have shown this previously). This must be discussed at length in the Discussion section, because it opens up fundamentally different interpretations of their findings (i.e. that, rather than general translocation of gut bacteria to the brain, a small number of opportunistic pathogens are able to access the brain via the vagus nerve given the opportunity afforded when the intestinal barrier is compromised). As stated previously, this is still very interesting, and could have significant impacts for health, but it is not the same as saying "gut bacteria" in general translocate. It also builds on existing data showing that some pathogens (mostly viruses, but also perhaps Borrelia species) are already known to be able to migrate along the vagus nerve. This should also be discussed.

3. Many of the other original points raised after the first round of peer review remain unaddressed. I have added additional information on these in the comments below.

Comments arising from the rebuttal letter:

Reviewer 1.

Previous major comment 2 - Many of the results are still presented in an anecdotal way, even after the text edits. Where results were not significant, this should be clearly stated in the main text. When numbers of translocated bacteria were very small (or even absent) this should also be clearly stated in the text. As discussed above, given the issues with the count data, many conclusions will need to be changed to being qualitative rather than quantitative, and this limitation should be discussed in the text.

Previous major comment 3 - Parts of this response should be added to the Discussion section. If issues are raised by peer reviewers, they are likely to be spotted by subsequent readers when the paper is published too.

Previous major comment 6 - I don't think just removing the word "commensal" has fully addressed this concern as I think many readers when they see "gut bacteria" will make the mental leap to this being applicable to all gut bacteria when, as explained above, many of these brain invading species (e.g. S. xylosus) are opportunistic pathogens, and not thought of as common gut bacteria. I therefore recommend that all uses of "gut bacterial translocation to the brain" throughout the text (including in key sections like the Abstract) be changed to something like "bacterial translocation from the gut to the brain". This is a subtle, but important, distinction.

Previous major comment 7 - As discussed above, the revised Discussion still lacks important critical assessment. It doesn't discuss previous evidence for pathogens invading via the vagus nerve, it doesn't explain how the Paigen diet compares to common human diets, the new section on human autopsies focusses on why bacteria might have been missed previously, rather than include the possibility that bacteria aren't actually present in the brains of most humans, and that results may not translate from these quite contrived mouse models to the situation in humans. It is all very one-sided, and lacking critical balance.

Previous specific comments line 1 - I strongly disagree with this response. The title is very open to mis- or over-interpretation, particularly by casual/skim readers, and must be changed to be more reflective of the actual findings of the study. It is clear that the authors want to drum up as much attention as possible, which is perfectly understandable, but doing so by over-extrapolating is not scientifically robust, or appropriate. The human microbiome field has been littered with numerous overblown concepts (e.g. links to obesity, the placental microbiome etc) that have subsequently been proven to be wrong or overblown, but only after tens of millions of dollars of taxpayers' money, and countless researcher time, has been spent chasing overhyped initial claims.

Previous specific comments lines 129 to 131 (and also lines 139 to 142 and lines 159 to 160) - These edited passages of text still lead the reader towards the authors' chosen hypothesis. The results are not significant. At the very least this caveat should be added to the main text.

Previous specific comments line 139 (and also lines 272 to 274) - Key parts of these responses should be added to the main text, so that all readers can see them?

Previous specific comments lines 172 to 172 - this point was only partially rebutted, what about the other factors listed in my previous query?

Previous specific comments lines 222 to 238 (and also lines 334 to 349) - I don't think that these queries are fully addressed, sorry. The authors state what they did, but do not provide data to show that cage effects have actually been eliminated, or at least factored into the subsequent analysis? It is clear across all of the results that there is significant variability, including many cases where zero bacteria were detected in mice brains within an experimental group, but were detected in others in the same group. It is important to know if any of that can be related to cage/batch effects.

Previous specific comments line 432 - thank you for correcting some of the inaccurate terminology, however, a number of uses are still uncorrected, e.g. lines 422, 760, 761, 762, 766, 809, 1025.

Previous specific comments line 460 - please add a reference (https://pmc.ncbi.nlm.nih.gov/articles/PMC3218848/) for the actual software to give the inventors of LEfSe appropriate credit.

Previous specific comments S2 Fig (and also Table S1, Table S3, Table S4) - perhaps the authors could fully clarify in their next response letter why the original version stated 3 independent experiments when only 2 were carried out, and why numbers in tables didn't agree with those in the figures? Apologies for the pushing this point, but it is important to understand where these discrepancies arose from.

Reviewer 2:

It may be good to add key text from the responses to "Dynamics of bacterial presence in the brain", and the results from the imaging of vagus nerves, to the main text. The latter in particular is important for making clear that the numbers of bacteria are very low.

Reviewer 3:

See comment on Reviewer 2 above - the imaging results and rebuttal text on low biomass should probably be added to the main text, not just included in the response letter?

Key text from the response to the point on "Temporal dynamics and localization in the brain" would be valuable additions to the main text. The response to the point on "Extension to neurological disease models" is a valuable limitation to add to the main text too?

Other specific comments, in order they appeared in the revised version of the main text (please note that line numbers are based on the track changes version of the text):

Line 1 - as discussed above, please change the title.

Line 52 - effect size is not captured anywhere in the Abstract. Please add "small numbers" or something like this here or in other parts of this section.

Line 58, and also lines 534 to 535 at the end of the Discussion - there is no proof at this point that the bacteria could play a role in diverse neurological conditions, or how knowledge of them will have profound transformative impact. The number of bacteria is typically very low, the authors show that translocation is reversible, the impact of even long-term colonisation is currently unknown, and they currently do not present any evidence for health impacts in either mice or humans (I acknowledge that this will be dealt with in later manuscripts). Therefore, at most, these current presented results indicate that further investigation is warranted. I think these two sentences at the end of the Abstract and Discussion should therefore be tempered to avoid over-interpretation.

Lines 74 to 90 - the reporting here is selective and one-sided. Reproducibility/consistency in human microbiome studies is typically very poor, and there are no accepted biomarkers of dysbiosis in neurodegenerative conditions. The presented findings are not consistent enough to be of clinical or diagnostic significance. This is not how they are currently presented though. Please add more balance to this section.

Lines 96 to 97 - I think something along the lines of "we demonstrate that small numbers of bacteria can translocate directly from the gut to the brain via the vagus nerve" would be more representative of what was found. Please see earlier comments regarding the potentially misleading use of "gut bacteria" throughout the rest of the manuscript from this point onwards too.

Line 101 - "suggesting" is too strong based on the level of evidence presented in this manuscript. Please see earlier (similar) comment about Line 58.

Line 104 - Similar to the title, I think this sub-heading is too broad.

Lines 115 to 116 (and all other similar uses) - how is "dysbiosis" defined and proven? For example, is it "bad" that Bacteroides and Akkermansia are enriched? I recommend the authors read this vitally important perspective on the overuse of the word https://www.nature.com/articles/nmicrobiol2016228, and perhaps edit the text in light of the limitations of this concept?

Figure 1E - looking at the new supplementary colony count file, it becomes clear that numbers are too low in most cases for accurate calculation of CFU/g (see earlier comment) but also, importantly, that the vast majority of detected colonies in the brains of these mice were S. xylosus. Since this is an opportunistic pathogen with previously demonstrated invasion of brain tissue in KO mice (see earlier comment) are the authors actually observing active infections by a specific set of capable pathogens rather than incidental translocation by gut bacteria in general? This is important for how the data are interpreted later.

Line 213 - to avoid potential over-interpretation by new readers, perhaps delete the word "microbiome" here? It is not necessary to include this word in order to make the point that the gut is the source of subsequent invasion of the vagus nerves.

Lines 220 to 221 - Figs 2A and 2B show total bacterial CFUs, not S. xylosus and E. faecalis?

Figure 2, and lines 294 to 296 - the figure labelling here doesn't match that given in the tabs of the supplementary colony counts spreadsheet? The tabs say "2G Ileum" and "2G Fecal samples", which doesn't match the figure labelling.

Line 329 - rather than put this text in a Supplemental Discussion most people will not read, I think it would be better to put some additional key points here in the main text, particularly the point about cell numbers never reaching high enough densities to be detected by standard PCR. This might also be a good point to add that you were unable to visualise cells using microscopy either? This all helps to underline that the invasion is typically very low-level, which is potentially important from an effect size point of view, but may also help to explain why this phenomenon has not been reported before?

Data in colony count Excel table tab for Figure 5E - I wondered why "Lactobacillus acidophilus/gasseri" was included as a column in the table but seemed to have no colonies? I also wondered why "Staphylococcus sp." was the header rather than "Staphylococcus xylosus". Was it a different species in these mice?

Line 417 - please add the word "mouse" before "models" here.

Line 522 - add "small numbers of" before "bacteria" to help emphasize typical effect size?

Line 540 - it states here cage effects were minimized. It explains how they tried to do this, but not how the success of this approach was assessed? Did the authors do statistical analyses to detect or definitively rule out any inter-cage variability within the overall results?

Lines 580 to 581 - the three uses of "poops" here could probably be changed to the more scientific "feces"? Furthermore, I wondered why the authors chose to dry the sample at 37oC rather than a higher temperature? Would this not encourage bacterial growth during the 48 hour drying process, and therefore potentially impact final measured biomass?

Lines 623 to 624 - this could perhaps be made a little clearer for new readers by showing the actual calculation that was used?

Lines 624 to 627 - it states that all colonies were identified by MALDI-ToF. Looking at the colony count data (bearing in mind it is incomplete, missing data from a number of figures, and won't include colonies that matched with different species from the ones listed in the tables) there are tens of thousands of colonies listed across all of the spreadsheet tabs. Did the authors really identify all of these with MALDI-ToF? This seems like a huge amount of work, and probably very expensive? Fair play if they did, but I thought it best to double check.

Lines 662 to 663, and line 727 - what "standard bacterial culture method" was used? There is not enough information here to allow independent replication.

Lines 706 to 707 - I presume uninoculated Mueller-Hinton Broth was included as a control? If so, this could be stated, for clarity?

Lines 731 to 732 - "various organs" is perhaps a little vague? Which organs? Similar, what PCR method/conditions were used? More information is needed for independent replication purposes?

Reviewer #2:

The authors have adequately addressed the issues that were raised.

Reviewer #3 (Archita Mishra):

The authors have addressed all my previous comments satisfactorily, and the manuscript has significantly improved in clarity and scientific rigour. I am now satisfied with the responses and revisions and recommend acceptance of the manuscript.

---

## [Decision Letter · Decision Letter 3]

16 Jan 2026

Dear Arash, dear Dave,

Thank you for your patience while we considered your revised manuscript "Translocation of bacteria from the gut to the brain" for publication as a Discovery Report at PLOS Biology. This revised version of your manuscript has been evaluated by the PLOS Biology editors, the Academic Editor and Reviewer 1.

Based on the reviews, we are likely to accept this manuscript for publication, provided you satisfactorily address ALL the remaining points raised by the reviewer 1. We are sure the paper will be under scrutiny once it is published, and therefore we think it is important to phare things carefully. Please also make sure to address the following data and other policy-related requests.

1) We routinely suggest changes to titles to ensure maximum accessibility for a broad, non-specialist readership, and to ensure they reflect the contents of the paper. In this case, we would suggest a minor edit to the title, as follows. Please ensure you change both the manuscript file and the online submission system, as they need to match for final acceptance:

"Translocation of bacteria from the gut to the brain occurs in mice fed a high-fat diet."

2) Unfortunately, Discovery Reports are limited to 4 main figures; currently you have 6 main figures. Please either combine them or send 2 to the supplementary materials to reduce the number to 4.

3) Please add weblink of the funding agencies in the Financial Disclosure statement in the main text as well as on the manuscript details during resubmission.

Please supply the numerical values either in the a supplementary file or as a permanent DOI’d deposition for the following figures:

Figure 1B-L, 2A-K, 3A-E, 5A-E, 6B-E, S1AB, S2A-I, S3A-H, S5AC-F, S6A-L, S7A-H, S8AB, S9, S10ABC, S11AB, S12ABDE, S15B, S16A, S17A, S18A, S19A-E. Table S6

5) Please cite the location of the data clearly in all relevant main and supplementary Figure legends, e.g. “The data underlying this Figure can be found in S1 Data” or “The data underlying this Figure can be found in https://doi.org/10.5281/zenodo.XXXXX”

6) Please provide the tree files for the phylogenetic trees in Figures 1A, 6A, S16B, S17B, S18B.

7) Supplementary files (e.g., excel). Please ensure that all data files are uploaded as 'Supporting Information' and are invariably referred to (in the manuscript, figure legends, and the Description field when uploading your files) using the following format verbatim: S1 Data, S2 Data, etc. Multiple panels of a single or even several figures can be included as multiple sheets in one excel file that is saved using exactly the following convention: S1_Data.xlsx (using an underscore).

8) Please ensure that your Data Statement in the submission system accurately describes where your data can be found and is in final format, as it will be published as written there

9) Per journal policy, if you have generated any custom code during the course of this investigation, please make it available without restrictions. Please ensure that the code is sufficiently well documented and reusable, and that your Data Statement in the Editorial Manager submission system accurately describes where your code can be found. More information on our Code Policy, what and how to share can be found here: https://journals.plos.org/plosbiology/s/code-availability

We expect to receive your revised manuscript within two weeks.

*Published Peer Review History*

*Press*

Sincerely,

Melissa

Melissa Vazquez Hernandez, Ph.D.

Associate Editor

PLOS Biology

REVIEWERS' COMMENTS

Reviewer #1:

I thank the authors for their consideration of my previous comments, and for the further edits to the manuscript. They have satisfactorily addressed many of them, and I also thought that the new "FAQs" document was a nice addition.

However, a number of the previous comments still remain unaddressed, or are only partially addressed. To be clear, I like this study, and I am supportive of it being published. However, the authors need to be more objective, and more realistic about what they can claim at this point in time with the data presented in this particular paper. As mentioned in my previous comments, overhype and overextrapolation have led to a huge amount of wasted time, funding and effort in the microbiome field, and this paper does not need to do it. With some further text edits, this will make an excellent addition to the literature. I hope my further (and hopefully last!) set of comments will prove useful.

Comments on the new response letter:

Previous major comments 2 and 6 - these remain unaddressed. Although the authors point out that they "do not agree that all the bacterial species found in the brain are always opportunistic pathogens", they give the example of lactobacilli, which are in fact only found as a relatively small proportion of their brain culture isolates, and in any case can also be opportunistic pathogens in compromised hosts - see https://pmc.ncbi.nlm.nih.gov/articles/PMC6560513/ for example). So, it is clear that the majority of the bacteria they are detecting in the brain are opportunistic pathogens with previously demonstrated virulence. The authors therefore need to point out somewhere in the Discussion section that only a very small subset, involving only certain types of bacteria out of the thousands of bacterial species that are present, end up in the brain (their text from later in the response letter on this point is good, and so could be cut and pasted into an appropriate point in the main text?), and that most of them have been shown previously to be opportunistic pathogens. It is also fundamentally important to point out that species like S. xylosus have been shown previously to invade the brains of KO mouse models (see previous peer review comments). This must be discussed at length in the Discussion section, because it opens up fundamentally different interpretations of their findings (i.e. that, rather than phylogenetically broad translocation of gut bacteria to the brain, a small number of (mostly) opportunistic pathogens are able to access the brain via the vagus nerve given the opportunity afforded when the intestinal barrier is compromised). As stated previously, this is still very interesting and novel, and could have significant impacts for health.

Title - I thank the authors for revising this, but it is still too non-specific. They need to add "in mice" or "murine". The work was done in mice, and that is the only host organism they present evidence for vagus nerve-based translocation occurring in. This is not an unreasonable request, and easily done. This minor edit also minimises the risk of skim readers glancing at the title, over-interpreting the findings and over-extrapolating to humans. Similar translocation might also happen in humans, but the authors do not provide evidence for this yet.

Cage effects - I thank the authors for clarifying in the response letter. The response was convincing, and helps provide some assurance that cage effects are not playing an important role here. However, I may not be the only reader to query this possibility, so they may wish to add some of the text from the response letter to the Methods or Discussion of the main text too?

Erroneous use of "16S rRNA" - thank you for correcting most of the previous uses, but some still remain (e.g. lines 606, 607, 878, 879 and 883).

Regarding my previous comment "perhaps the authors could fully clarify in their next response letter why the original version stated 3 independent experiments when only 2 were carried out, and why numbers in tables didn't agree with those in the figures? Apologies for the pushing this point, but it is important to understand where these discrepancies arose from" - this is not addressed. The authors state that they did lots of experiments, but chose to only show representative data. The response letter does not explain why tables and figures were generated with different subsets of data rather than the full dataset, or explain the rationale for only selectively showing representative data rather than the full dataset.

Regarding my previous comment on over-interpretation in the final lines of the Abstract and the Discussion - this remains unaddressed. Phrases like "offers hope", "potentially have a profound, transformative impact" and "elucidate how this phenomenon" are too emotive or indicate a level of surety that the phenomenon definitely occurs in humans that is unsupported by the currently presented dataset. Patient groups will likely read this work. It is not fair to them to over-hype when the authors have not yet even demonstrated that the same phenomenon occurs in humans (at least in the current manuscript). I have made further suggestions in the "Specific comments on the main text" section below to temper the language.

Regarding the response "We have now revised this section removing the discussion of biomarkers of dysbiosis in neurological conditions and bringing more balance to this section" - I'm afraid the revised text does little to address the previous comments. The corresponding section in the main text introduction is still not balanced, at all.

I presume the authors will have read this recent article, but if not, I strongly recommend they take on board the main conclusions:

https://www.cell.com/neuron/fulltext/S0896-6273(25)00785-8

Figure 3 in that manuscript is highly illustrative of similar poor reproducibility found with basically all attempts to link the microbiome with disease. Much of the research that has been done attempting to correlate microbiome with neurological conditions is poorly performed, and there are few/no consistent patterns recorded across studies.

That is not at all how the current revised main text reads. There is no mention of inconsistencies between studies. It doesn't state that, overall, there are no consistent biomarkers of "dysbiosis" in these conditions. That is what should be written in this section, rather than just cherry picking and listing some taxa again. Please see the "Specific comments on the main text" section below for further comments.

Regarding my previous query starting with "Line 329" - it may be good for the authors to include the microscopy based results in the Results section (perhaps next to the initial PCR section?). This is mentioned in the Discussion of the main text, but appears a little out of the blue and without context there as it was not described earlier on in the Results section?

Regarding the response "we occasionally detect Staphylococcus species other than S. xylosus, such as S. lentus or S. sciuri. In one of the mice in question, we detected S. lentus instead of S. xylosus. Therefore, we combined the colony counts for these species under Staphylococcus sp." - Thanks for clarifying this point. To make this clearer for all new readers, the authors could perhaps add this info to the legend, or as a footnote at the bottom of the relevant Excel table tabs?

Regarding the response to my previous query about classifying colonies using MALDI-ToF - I thank the authors for providing clarity on this point. However, I'm afraid the response indicates that these findings may not be robust. Picking "one morphologically similar colony of each type" for MALDI-ToF is not ideal since different bacteria types can very commonly look quite morphologically similar as colonies on plates (particularly if they are small). There is therefore no guarantee that all of these colonies would be the same species, and so the authors cannot be sure that their species-level count data are accurate. At the very least, a caveat should be added to the Methods or Discussion section making it clear that the individual species count data are estimates, and not based on empirical data. Please also see comment "Lines 732 to 736" below.

Specific comments on the main text, in order they occurred (please note that line numbers refer to the track changes version):

Lines 44 to 45 - "strong correlation between gut dysbiosis and neurological diseases" is an over-extrapolation. As discussed above, although differences in microbiome composition are often reported, they are rarely consistent. This weakens the argument for a "strong" correlation. The use of "dysbiosis" on line 94 is also not warranted, given the complete lack of clinical relevance of this term due to huge inconsistencies between studies and a lack of reproducible biomarkers that define this microbiome state in disease.

Line 61 - as discussed above, "….further investigation to elucidate how this phenomenon may play a role…." is too definitive, as the "how" makes the assumption that it definitely does play a role in humans, it is only the mechanism that needs to be elucidated. A more accurate summing up sentence would be something like, "…further investigation to determine if this phenomenon also occurs in humans, and elucidate whether or not it may play a role…"

Lines 78 to 79 - As discussed above, this is too vague, and unbalanced. Have these taxa consistently been seen to be altered across all studies of the gut microbiome and AD? If not, state that here.

Lines 80 to 81, and also lines 83 to 84, and 87 - these sentences are very vague. How so? What putative mechanisms are involved?

Lines 85 to 86 - Perhaps state why is this interesting? This is unlikely to be clear to all new readers? Furthermore, are Prevotellaceae always "markedly diminished" in Parkinson's patients? Has this been reproduced consistently across many studies? If not, the authors should state this rather than being so definitive here. Please see earlier comments about avoiding cherry picking results from individual studies, particularly if they are not reproduced in other studies.

Lines 88 to 93 - please see this paper: https://www.cell.com/neuron/fulltext/S0896-6273(25)00785-8. The evidence for consistent links between specific microbiome taxa and autism is very poor. Rather than just cherry picking taxa, this section needs to be re-written to reflect this. Similarly, "Clostridia" is an entire class of bacteria containing a hugely diverse group of hundreds of constituent species, with very different activities and metabolite profiles. It is far too overly-simplistic to claim this entire taxon is playing a key role in autism or releasing neurotoxins.

Lines 104 to 105 - to temper over-extrapolation, please change "…raising the possibility that this pathophysiological pathway may contribute to…" to something more like "…raising the possibility that, if this pathophysiological pathway also occurs in humans, it may contribute to…".

Lines 281 to 283 - to temper over-extrapolation, please change the text to something more like, "Although results were not statistically significant, these temporal data provide further anecdotal support for the vagus nerve......."

Line 472 - here might be a good place to specifically point out that only a subset of bacteria seem to be capable of translocating?

Line 592 - are there some structural issues with this sentence? Could be edited to improve clarity?

Line 605 - some new readers may not understand why lack of negative controls makes results challenging to interpret? Add something like, "due to the well-established issue of contamination impacting low biomass microbiome studies" to improve clarity?

Line 626 - remove "that" here, since it is already used in the preceding line?

Lines 637 to 640 - Add something like "In the event that this phenomenon also occurs in humans," before "further studies". I think "and potentially have a profound, transformative impact" should be deleted here too. It overhypes results presented here (which are based on rather contrived mouse models, with no evidence presented yet for the same phenomenon happening in humans) and potentially gives false short-term hope to patient groups.

Lines 732 to 736 - this may be a good place to add the caveating point I mentioned above about the potentially flawed nature of this single "representative colony morphology" based approach?

Line 928 - I checked the SRA website, but this data didn't seem to be available? Presumably it is still under embargo. Please can the authors ensure it is released after acceptance of the manuscript so that others can assess the data themselves should they so wish.

Supplemental Figures S13B, S13C, S14 and S20 - it was unclear to me why some of the bands are colored red. Perhaps the authors can add some explanatory text to the figure legends?

---

## [Editor Report · Decision Letter 4]

27 Jan 2026

Dear Arash, dear Dave,

Thank you for the submission of your revised Discovery Report "Translocation of bacteria from the gut to the brain in mice" for publication in PLOS Biology. It is my pleasure to say on behalf of my colleagues and the Academic Editor, Ken Cadwell, that we can in principle accept your manuscript for publication, provided you address any remaining formatting and reporting issues. These will be detailed in an email you should receive within 2-3 business days from our colleagues in the journal operations team; no action is required from you until then. Please note that we will not be able to formally accept your manuscript and schedule it for publication until you have completed any requested changes.

Thank you for agreeing to add mice into the title. It is journal's policy that, when the results have implications for human disease (as in this case Alzheimer’s, Parkinson’s, and autism spectrum disorder), if the experiments were done only in animal models, we TRY to include the model in the title. This is because there is research to show that otherwise, press coverage is misleading and indeed offers unsubstantiated hope to patients. The work is important as it is and will be influential, and we want to avoid the risk of overhyping.

*IMPORTANT*: because we think this will be an influential piece of work, we would like to encourage you to opt in into Transparent Peer Review.

PRESS

Sincerely,

Melissa

Melissa Vazquez Hernandez, Ph.D., Ph.D.

Associate Editor

PLOS Biology
